**Data Availability Statement:** Data cannot be shared publicly because the PATOS (Pan-Asian Trauma Outcomes Study) data are managed by the

# Association between prehospital field to emergency department delta shock index and in-hospital mortality in patients with torso and extremity trauma: A multinational, observational study

**Dae Kon Kim**[1,2], **Joo Jeong**[1,2]*, **Sang Do Shin**[2,3], **Kyoung Jun Song**[2,4], **Ki Jeong Hong**[2,5], **Young Sun Ro**[2,5], **Tae Han Kim**[2,4], **Sabariah Faizah Jamaluddin**[6], **for the PATOS Clinical Research Network**¶

1 Department of Emergency Medicine, Seoul National University Bundang Hospital, Seongnam, South Korea, 2 Laboratory of Emergency Medical Services, Seoul National University Hospital Biomedical Research Institute, Seoul, South Korea, 3 Department of Emergency Medicine, Seoul National University College of Medicine, Seoul, South Korea, 4 Department of Emergency Medicine, Seoul National University Boramae Medical Center, Seoul, South Korea, 5 Department of Emergency Medicine, Seoul National University Hospital, Seoul, South Korea, 6 Department of Emergency Medicine, Universiti Teknologi MARA Sungai Buloh, Selangor, Malaysia

¶ Investigators are listed in S1 Appendix.
* joojeong@snubh.org

## Abstract

Hemorrhage, a main cause of mortality in patients with trauma, affects vital signs such as blood pressure and heart rate. Shock index (SI), calculated as heart rate divided by systolic blood pressure, is widely used to estimate the shock status of patients with hemorrhage. The difference in SI between the emergency department and prehospital field can indirectly reflect urgency after trauma. We aimed to determine the association between delta SI (DSI) and in-hospital mortality in patients with torso or extremity trauma. Patients with DSI >0.1 are expected to be associated with high mortality. This retrospective, observational study used data from the Pan-Asian Trauma Outcomes Study. Patients aged 18–85 years with abdomen, chest, upper extremity, lower extremity, or external injury location were included. Patients from China, Indonesia, Japan, Philippines, Thailand, and Vietnam; those who were transferred from another facility; those who were transferred without the use of emergency medical service; those with prehospital cardiac arrest; those with unknown exposure and outcomes were excluded. The exposure and primary outcome were DSI and in-hospital mortality, respectively. The secondary and tertiary outcome was intensive care unit (ICU) admission and massive transfusion, respectively. Multivariate logistic regression analysis was performed to test the association between DSI and outcome. In total, 21,534 patients were enrolled according to the inclusion and exclusion criteria. There were 3,033 patients with DSI >0.1. The in-hospital mortality rate in the DSI >0.1 and ≤0.1 groups was 2.0% and 0.8%, respectively. In multivariate logistic regression analysis, the DSI ≤0.1 group was considered the reference group. The unadjusted and adjusted odds ratios of in-hospital

PATOS CRN. Data are available from the PATOS CRN for researchers who meet the criteria for access to confidential data (http://lems.re.kr/eng/patos-crn-faq). The data underlying the results presented in the study are available from PATOS CRN (patos.crn@gmail.com).

**Funding:** The author(s) received no specific funding for this work.

**Competing interests:** The authors have declared that no competing interests exist.

mortality in the DSI >0.1 group were 2.54 (95% confidence interval [CI] 1.88–3.42) and 2.82 (95% CI 2.08–3.84), respectively. The urgency of traumatic hemorrhage can be determined using DSI, which can help hospital staff to provide proper trauma management, such as early trauma surgery or embolization.

## Introduction

Trauma is the leading cause of morbidity and mortality in all age groups. Over the past decade, the rate of mortality due to trauma has increased up to 23% [1]. Hemorrhage is one of the most important causes of mortality in preventable deaths after trauma [2]. In addition, trauma deaths after hospital admission are usually related to massive hemorrhage, which can be preventable if detected early by hospital staff [3]. Hemorrhage causes hypovolemic shock, compounded by lactic acidosis, hypothermia, and coagulopathy. Shock status must be corrected by hemostasis via embolization or emergency laparotomy and transfusion therapy [4].

Vital signs are one of the most important tools used to indirectly evaluate a patient's status. In hypovolemic shock, the degree of shock severity is classified by heart rate (HR), systolic blood pressure (SBP), and Glasgow Coma Scale score [5]. Because individual vital signs alone cannot accurately predict outcomes, shock index (SI), calculated as HR divided by SBP, has been developed to evaluate shock status in diverse conditions [6–8]. Recently, delta SI (DSI), which is the change in SI over time, has been developed to assess shock severity, high-risk patients for massive transfusion, and mortality [9–12]. In previous studies, a DSI of 0.1–0.3 has been related to worse outcomes [9].

Even if patients have a similar severity of trauma, the cascade of physiological deterioration is relatively different according to the anatomical location of the trauma. The main causes of deterioration are brain herniation in traumatic brain injury (TBI), respiratory compromise due to upper airway bleeding in pan-facial injury, and hypovolemic shock in torso and extremity injury. Therefore, a treatment plan should be developed according to the anatomical location and severity of the injury. However, in previous studies on DSI, all anatomical lesions, including traumatic brain, facial, and neck injuries, were included in the analysis [9, 10]. Because vital signs represent the severity of hypovolemic shock, it is reasonable to adapt vital sign parameters in anatomical lesions that are highly related to hypovolemic hemorrhage.

Therefore, this study aimed to determine the association between DSI, from the prehospital field to the emergency department (ED), and in-hospital mortality in patients with torso and extremity injuries.

## Methods

### Study design and setting

This retrospective, international, and cross-sectional study used data from the Pan-Asian Trauma Outcomes Study (PATOS).

### Data source and collection

The PATOS is a registry of trauma cases from participating hospitals across the Asia-Pacific countries. It was established in 2013 to collect trauma data from the Asia-Pacific region. All participating hospitals have standardized definitions of variables by adopting a consensual common taxonomy and data collection methodology. Patients with trauma who are

transported to the ED of the participating hospitals via typical emergency medical services (EMS) ambulances in developed countries or other types of ambulances in developing countries are included in the PATOS. The PATOS collects information on demographic findings, injury epidemiology, prehospital factors, hospital factors, and outcomes of patients with injury. Prehospital data are collected from ambulance run sheets or EMS dispatch records. Hospital records and patient outcome data are collected from the hospital medical records. To maintain standardized and consistent data quality, training modules were developed to educate all personnel involved in registering data. All data are enrolled via an electronic data capture system. The PATOS Data Quality Management Committee (QMC) monitors invalid and/or incomplete data forms and provides feedback to each participating hospital. All hospitals respond to the PATOS Data QMC reports within 2 weeks for data correction [13–15].

## Study population

All PATOS cases from January 2015 to November 2018 were initially enrolled in the analysis. Patients from China, Indonesia, Japan, Philippines, Thailand, and Vietnam were excluded because essential variables were investigated in a small number of patients. Patients aged <18 years or >85 years; those with prehospital cardiac arrest; those transferred from another hospital; those who were transferred without the use of EMS; those with anatomical injury in the head, face, neck, and spine; those with unknown prehospital and ED SBP or HR; those with outlying SBP or HR; and those with unknown outcomes were excluded.

## Exposure and outcome variables

The exposure was defined as DSI, i.e., the first EMS SI was subtracted from the ED SI. The DSI cutoff value of 0.1 was used according to the values used in previous studies [9, 10]. The initial SBP and HR measured at the prehospital and ED visits were used to calculate the prehospital and ED SI. Prehospital SI was calculated by dividing prehospital HR by prehospital SBP. ED SI was calculated by dividing ED HR by ED SBP.

The following data were extracted from the database: demographics (age, gender, country, mechanism of injury, and intent of injury), prehospital information (EMS time, prehospital SBP, prehospital HR, and prehospital SI), and hospital information (ED SBP, ED HR, ED SI, anatomical location of injury, ISS, and length of stay [LOS, days] in the intensive care unit [ICU]). An ISS of 9–15 indicated moderate injury severity, while an ISS >15 indicated severe injury.

The primary outcome was in-hospital mortality. The secondary outcome was ICU admission. The tertiary outcome was massive transfusion during hospital admission. The massive transfusion was defined as total transfusion amount more than 4,000ml within 24 hours after ED admission. In addition, embolization and surgery were also analyzed. Embolization and surgery were defined as those performed on the thorax, abdomen, upper extremity, and lower extremities.

## Statistical analyses

Patient demographic factors, such as age, gender, country, EMS time, mechanism of injury, intent of injury, anatomical location of injury, prehospital and ED SBP and HR, ISS, ICU LOS, and outcomes, were compared according to the DSI cutoff value. Categorical variables, presented as numbers and percentages, were compared using the chi-square test. Continuous variables, presented as median and interquartile range (IQR), were compared using the Wilcoxon rank-sum test. We performed multivariate logistic regression analysis to test the association between DSI and outcomes. The DSI ≤0.1 group was used as the reference group in the

analysis. Potential confounders, such as age, gender, country, EMS time, mechanism of injury, intent of injury, and anatomical location of injury were adjusted. In a massive transfusion, embolization, and surgery, the country was excluded from the confounding variable because few patients were in a specific country (Taiwan). Adjusted odds ratios (AORs) and 95% confidence intervals (CIs) were calculated for the outcomes. Subgroup analysis was performed to compare the effect of prehospital SI (SI $\leq$0.9 and SI >0.9) on the outcomes. All analyses were performed using the Statistical Analysis System version 9.4 (SAS© Cary, NC, USA).

### Ethics statement

The Institutional Review Boards (IRBs) of the hospitals of the PATOS Clinical Research Network approved this study (IRB No. 1509-045-702) and waived the requirement of patient consent.

### Results

Of the 71,383 patients included in the PATOS, 21,534 patients were finally analyzed, excluding patients from China, Indonesia, Japan, Philippines, Thailand, and Vietnam (N = 15,225); those aged <18 years or >85 years (N = 8,829), those with prehospital cardiac arrest (N = 575); those transferred from another facility or those who were transferred without the use of EMS (N = 6,501); those with injury in the head, face, neck, spine, or body surface (N = 13,444); those with unknown or outlying prehospital and ED SBP or HR (N = 3,725), and those with unknown outcomes (N = 1,550) (Fig 1).

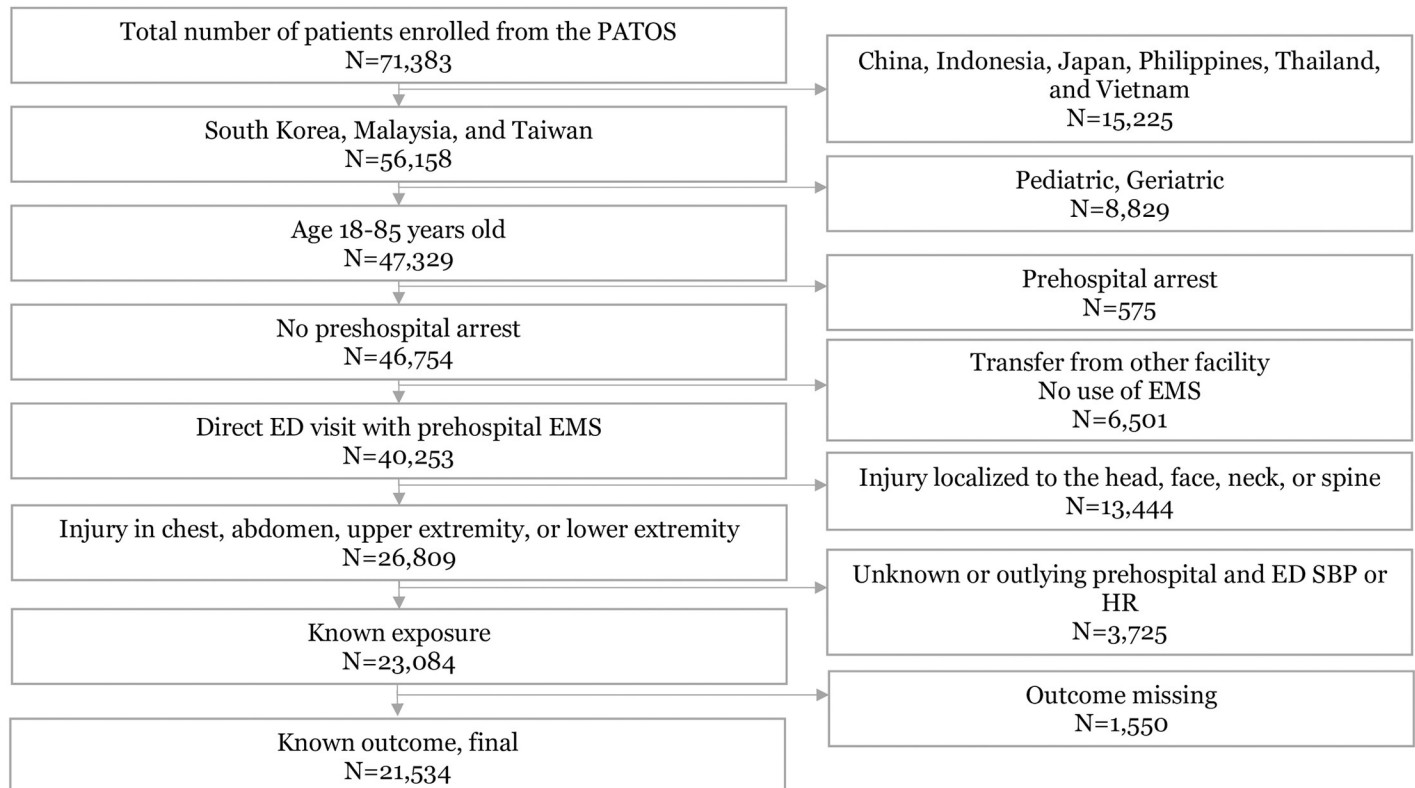

PATOS, Pan-Asian Trauma Outcomes Study; ED, Emergency Department; EMS, Emergency Medical Services; SBP, Systolic Blood Pressure; HR, Heart Rate.

**Fig 1. Study flowchart.** Abbreviations: PATOS, Pan-Asian Trauma Outcomes Study; ED, emergency department; EMS, emergency medical services; ISS, Injury Severity Score.

**Table 1. Demographic findings according to exposure groups.**

|  |  | Total | DSI ≤0.1 | DSI >0.1 | P value |
|---|---|---|---|---|---|
|  |  | N (%) | N (%) | N (%) |  |
|  |  | 21534 (100) | 18501 (100) | 3033 (100) |  |
| Age | Median (IQR) | 47 (29–64) | 48 (29–64) | 43 (28–60) | <0.01 |
| Sex | Male | 13247 (61.5) | 11342 (61.3) | 1905 (62.8) | 0.11 |
| Country |  |  |  |  |  |
|  | South Korea | 13857 (64.3) | 11783 (63.7) | 2074 (68.4) | <0.01 |
|  | Malaysia | 6428 (29.9) | 5559 (30.0) | 869 (28.6) |  |
|  | Taiwan | 1249 (5.8) | 1159 (6.3) | 90 (3.0) |  |
| EMS call to ED arrival time (min) |  |  |  |  |  |
|  | Median (IQR) | 36 (28–46) | 36 (28–46) | 36 (28–46) | 0.27 |
| Mechanism of injury |  |  |  |  |  |
|  | Blunt | 18042 (83.8) | 15620 (84.4) | 2422 (79.9) | <0.01 |
|  | Penetrating | 1064 (4.9) | 869 (4.7) | 195 (6.4) |  |
|  | Others | 2428 (11.3) | 2012 (10.9) | 416 (13.7) |  |
| Anatomical location associated with AIS score ≥3 |  |  |  |  |  |
|  | Chest | 628 (2.9) | 495 (2.7) | 133 (4.4) | <0.01 |
|  | Abdomen | 135 (0.6) | 102 (0.6) | 33 (1.1) | <0.01 |
|  | Upper extremity | 164 (0.8) | 143 (0.8) | 21 (0.7) | 0.64 |
|  | Lower extremity | 2121 (9.8) | 1833 (9.9) | 288 (9.5) | 0.48 |
| EMS SI | Median (IQR) | 0.64 (0.56–0.74) | 0.65 (0.57–0.75) | 0.6 (0.53–0.68) | <0.01 |
| ED SI | Median (IQR) | 0.61 (0.51–0.72) | 0.59 (0.5–0.68) | 0.8 (0.71–0.92) | <0.01 |
| EMS SBP | Median (IQR) | 130 (120–145) | 130 (120–145) | 132 (120–148) | <0.01 |
| EMS HR | Median (IQR) | 85 (76–95) | 85 (78–96) | 80 (72–89) | <0.01 |
| ED SBP | Median (IQR) | 138 (122–156) | 140 (126–158) | 119 (106–134) | <0.01 |
| ED HR | Median (IQR) | 84 (75–95) | 82 (73–92) | 97 (86–108) | <0.01 |

Abbreviations: DSI, delta shock index; IQR, interquartile range; EMS, emergency medical services; ED, emergency department; AIS, Abbreviated Injury Scale; SI, shock index; SBP, systolic blood pressure; HR, heart rate.

According to the DSI cutoff value, there were 18,501 (85.9%) patients in the DSI ≤0.1 group and 3,033 (14.1%) patients in the DSI >0.1 group. The most common mechanism of injury was blunt injury in both groups. The most common injury location with AIS score ≥3 was the lower extremity in both groups. The median prehospital SI was 0.65 (IQR, 0.57–0.75) and 0.6 (IQR, 0.53–0.68) in the DSI ≤0.1 and DSI >0.1 groups, respectively. The median ED SI was 0.59 (IQR, 0.5–0.68) and 0.8 (IQR, 0.71–0.92) in the DSI ≤0.1 and DSI >0.1 groups, respectively (Table 1).

The median ISS was 4, with an IQR of 1–5 and 1–6 in the DSI ≤0.1 and DSI >0.1 groups, respectively. The median ICU LOS was longer in the DSI >0.1 group than in the DSI ≤0.1 group (median [IQR]: 5 [2–11] vs. 3 [2–8] days). The rates of primary (in-hospital mortality), secondary (ICU admission), and tertiary outcomes (massive transfusion) in the DSI ≤0.1 and DSI >0.1 groups were 0.8% and 2.0%, 5.5% and 10.4%, and 0.2% and 0.9%, respectively. Embolization and surgery were also performed significantly more in the DSI > 0.1 group (Table 2).

In multivariate logistic regression analysis, compared with the DSI ≤0.1 group (reference), the AORs were 2.82 (95% CI, 2.08–3.84) for in-hospital mortality, 2.02 (95% CI, 1.76–2.32) for ICU admission, and 5.24 (95% CI, 3.10–8.85) for massive transfusion in the DSI >0.1 group (Table 3). For embolization and surgery, the DSI > 0.1 group showed a significantly higher AOR (S1 Table).

**Table 2. In-hospital information and outcomes according to exposure groups.**

| | | Total | DSI≤0.1 | DSI>0.1 | P-value |
|---|---|---|---|---|---|
| | | N (%) | N (%) | N (%) | |
| | | **21534 (100)** | **18501 (100)** | **3033 (100)** | |
| ISS | | | | | <0.01 |
| | 1–8 | 17687 (82.1) | 15246 (82.4) | 2441 (80.5) | |
| | 9–15 | 3023 (14.0) | 2586 (14.0) | 437 (14.4) | |
| | 16–24 | 637 (3.0) | 525 (2.8) | 112 (3.7) | |
| | 25- | 187 (0.9) | 144 (0.8) | 43 (1.4) | |
| ICU length of stay | | | | | <0.01 |
| | Median (IQR) | 4 (2–9) | 3 (2–8) | 5 (2–11) | |
| Location of embolization | | | | | |
| | Chest | 14 (0.1) | 9 (0.0) | 5 (0.2) | 0.02 |
| | Abdomen | 27 (0.1) | 14 (0.1) | 13 (0.4) | <0.01 |
| | Upper extremity | 8 (0.0) | 6 (0.0) | 2 (0.1) | 0.37 |
| | Lower extremity | 27 (0.1) | 20 (0.1) | 7 (0.2) | 0.08 |
| In-hospital mortality | | 213 (1.0) | 151 (0.8) | 62 (2.0) | <0.01 |
| ICU admission | | 1326 (6.2) | 1011 (5.5) | 315 (10.4) | <0.01 |
| Massive Transfusion | | 58 (0.3) | 31 (0.2) | 27 (0.9) | <0.01 |
| Embolization* | | 65 (0.3) | 43 (0.2) | 22 (0.7) | <0.01 |
| Surgery | | 1569 (7.3) | 1311 (7.1) | 258 (8.5) | <0.01 |

Abbreviations: DSI, delta shock index; ISS, Injury Severity Score; IQR, interquartile range; ICU, intensive care unit.

* The total may not match because patients have undergone embolization in two or more sites.

In the subgroup analysis, the median age was 36 (IQR, 24–51) years in the EMS SI >0.9 and DSI ≤0.1 groups. The proportion of ISS patients with a score of 16 or higher was the highest in the EMS SI > 0.9 and DSI > 0.1 group (17.5%). The in-hospital mortality rates in the EMS SI ≤0.9 and DSI ≤0.1 groups, EMS SI ≤0.9 and DSI >0.1 groups, EMS SI >0.9 and DSI ≤0.1 groups, and EMS SI >0.9 and DSI >0.1 groups were 0.5%, 1.5%, 4.5%, and 13.4%, respectively. The proportion of those who underwent embolization and surgery was significantly higher in the group with high SI or DSI (Table 4).

**Table 3. Association between exposure groups and outcomes in multivariate logistic regression.**

| | In-hospital mortality | |
|---|---|---|
| | **Unadjusted OR (95% CI)** | **Adjusted OR (95% CI)** * |
| DSI ≤0.1 | Reference | Reference |
| DSI >0.1 | 2.54 (1.88–3.42) | 2.82 (2.08–3.84) |
| | ICU admission | |
| DSI ≤0.1 | Reference | Reference |
| DSI >0.1 | 2.57 (2.01–3.29) | 2.02 (1.76–2.32) |
| | Massive transfusion | |
| DSI ≤0.1 | Reference | Reference |
| DSI >0.1 | 5.35 (3.19–8.98) | 5.24 (3.10–8.85) |

Abbreviations: OR, odds ratio; CI, confidence interval; DSI, delta shock index; ICU, intensive care unit; EMS, emergency medical services.

*Adjusted for age, sex, country, EMS time, mechanism of injury, intent of injury, location of injury (excluding country variable for massive transfusion).

**Table 4. Association between exposure groups and outcomes according to prehospital shock index in multivariate logistic regression analysis.**

| | | Total | EMS SI ≤0.9 | | EMS SI >0.9 | | P value |
|---|---|---|---|---|---|---|---|
| | | | DSI ≤0.1 | DSI >0.1 | DSI ≤0.1 | DSI>0.1 | |
| | | N (%) | N (%) | N (%) | N (%) | N (%) | |
| | | 21534 (100) | 16998 (100) | 2884 (100) | 1503 (100) | 149 (100) | |
| Age | Median (IQR) | 47 (29–64) | 49 (30–65) | 44 (28–60) | 36 (24–51) | 39 (29–56) | <0.01 |
| Sex | Male | 13247 (61.5) | 10394 (61.1) | 1810 (62.8) | 948 (63.1) | 95 (63.8) | 0.19 |
| EMS SI | Median (IQR) | 0.64 (0.56–0.74) | 0.63 (0.56–0.72) | 0.6 (0.53–0.67) | 1 (0.95–1.09) | 1.05 (0.94–1.17) | <0.01 |
| ED SI | Median (IQR) | 0.61 (0.51–0.72) | 0.58 (0.49–0.66) | 0.79 (0.7–0.9) | 0.79 (0.67–0.95) | 1.31 (1.2–1.59) | <0.01 |
| ISS | | | | | | | <0.01 |
| | 1–8 | 17687 (82.1) | 14112 (83.0) | 2351 (81.5) | 1134 (75.4) | 90 (60.4) | |
| | 9–15 | 3023 (14.0) | 2346 (13.8) | 404 (14.0) | 240 (16.0) | 33 (22.1) | |
| | 16–24 | 637 (3.0) | 440 (2.6) | 94 (3.3) | 85 (5.7) | 18 (12.1) | |
| | 25- | 187 (0.9) | 100 (0.6) | 35 (1.2) | 44 (2.9) | 8 (5.4) | |
| In-hospital mortality | | 213 (1.0) | 84 (0.5) | 42 (1.5) | 67 (4.5) | 20 (13.4) | <0.01 |
| ICU admission | | 1326 (6.2) | 789 (4.6) | 260 (9.0) | 222 (14.8) | 55 (36.9) | <0.01 |
| Massive transfusion | | 58 (0.3) | 18 (0.1) | 18 (0.6) | 13 (0.9) | 9 (6.0) | <0.01 |
| Embolization | | 65 (0.3) | 36 (0.2) | 17 (0.6) | 7 (0.5) | 5 (3.4) | <0.01 |
| Surgery | | 1569 (7.3) | 1146 (6.7) | 227 (7.9) | 165 (11.0) | 31 (20.8) | <0.01 |

Abbreviations: EMS, emergency medical services; SI, shock index; DSI, delta shock index; IQR, interquartile range; ED, emergency department; ISS, Injury Severity Score; ICU, intensive care unit.

In multivariate logistic regression analysis for subgroup analysis, compared with the EMS SI ≤0.9 and DSI ≤0.1 groups (reference), the AORs for in-hospital mortality were 3.45 (95% CI, 2.36–5.04) in the EMS SI ≤0.9 and DSI >0.1 groups, 10.5 (95% CI, 7.49–14.8) in the EMS SI >0.9 and DSI ≤0.1 groups, and 41.8 (95% CI, 24.1–72.5) in the EMS SI >0.9 and DSI >0.1 groups (Table 5). For embolization and surgery, the high SI or SDI group showed a significantly higher AOR (S2 Table).

## Discussion

The results of this study revealed that DSI >0.1 was associated with higher rates of mortality, ICU admission, and massive transfusion. This trend was the same regardless of the EMS SI status in the subgroup analysis. DSI can reflect hemodynamic changes in early phases without the need for radiologic imaging, such as computed tomography (CT) or ultrasound. This parameter is more useful than an individual vital sign alone or SI measured once, which cannot provide significant clinical information on the time trend. The significance of this study is that it provides clues to identify individuals with poor prognosis so that early preparation can be made as soon as vital signs are checked at the ED admission. This information will help clinicians decide whether to activate the trauma team or initiate transfusion.

SI is associated with hemodynamic instability [16]. SI consists of SBP and HR and is significantly easy to assess in any situation. An SI >0.9 is considered unstable [6]. However, a single evaluation of SI should be interpreted cautiously because the trend of vital signs is more important than fragmentary measurements of vital signs. DSI is the trend of vital signs with higher accuracy than SI at a single timepoint [8, 10]. DSI >0.1 suggests an increase in SI compared to that in the prehospital field and implies ongoing bleeding in internal organs if the wound is not observed from the outside. As shown in Tables 2 and 3, DSI >0.1 was associated with a higher rate of embolization, and the rate was significantly higher in the DSI >0.1 group

**Table 5. Association between exposure groups and outcomes according to prehospital shock index in multivariate logistic regression.**

| | | In-hospital mortality | |
| --- | --- | --- | --- |
| | | Unadjusted OR (95% CI) | Adjusted OR (95% CI)* |
| EMS SI ≤0.9 | DSI ≤0.1 | Reference | Reference |
| | DSI >0.1 | 2.98 (2.05–4.32) | 3.45 (2.36–5.04) |
| EMS SI >0.9 | DSI ≤0.1 | 9.40 (6.79–13.0) | 10.5 (7.49–14.8) |
| | DSI >0.1 | 31.2 (18.6–52.4) | 41.8 (24.1–72.5) |
| | | ICU admission | |
| EMS SI ≤0.9 | DSI ≤0.1 | Reference | Reference |
| | DSI >0.1 | 2.04 (1.76–2.36) | 2.07 (1.78–2.41) |
| EMS SI >0.9 | DSI ≤0.1 | 3.56 (3.04–4.18) | 3.23 (2.73–3.81) |
| | DSI >0.1 | 12.0 (8.55–16.9) | 12.3 (8.55–17.6) |
| | | Massive transfusion | |
| EMS SI ≤0.9 | DSI ≤0.1 | Reference | Reference |
| | DSI >0.1 | 5.93 (3.08–11.4) | 6.10 (3.15–11.8) |
| EMS SI >0.9 | DSI ≤0.1 | 8.23 (4.03–16.8) | 11.4 (5.45–23.7) |
| | DSI >0.1 | 60.6 (26.8–137.3) | 71.8 (30.7–168.2) |

Abbreviations: OR, odds ratio; CI, confidence interval; DSI, delta shock index; ICU, intensive care unit; EMS, emergency medical services.

*Adjusted for age, sex, country, EMS time, mechanism of injury, intent of injury, location of injury (excluding country variable for massive transfusion).

than in the DSI ≤0.1 group. Therefore, the DSI is >0.1 at ED admission, medical staff should be aware of ongoing hemorrhage and should prepare for hemostasis and transfusion from the early treatment phase. The effect of DSI is more synergetic when used with SI. Individuals with prehospital SI ≥0.9 and DSI >0.1 must be treated with the highest priority because these values suggest the highest rate of mortality, ICU admission, and embolization (Table 4).

Schellengerg et al. analyzed DSI in patients with trauma, excluding the Cushing response due to terminal herniation and neurogenic shock [9]. Although they analyzed DSI between ED arrival and departure, the results was similar to those reported in our study, showing higher mortality and ICU LOS in the DSI >0.1 group. The cause of death was exclusively TBI, which accounted for 84% of all deaths. This result implies that different pathophysiologies according to anatomical regions should be considered, especially in TBI. The vital sign change in Cushing's triad, which comprises respiratory irregularity, widened pulse pressure, and bradycardia, has different pathophysiologies and requires different approaches and treatments for hypovolemic shock [17, 18]. Further studies dealing with vital signs in trauma must consider the pathophysiology of anatomical regions for different treatment plans.

The advantages of DSI can be highlighted in the prehospital stage. It is best to collect as much clinical information as possible, such as data on CT and ultrasound findings, hemoglobin level, and tissue hemoglobin oxygen saturation, to evaluate hemorrhage and shock status and determine diagnostic and treatment plans. Although Focused Assessment with Sonography in Trauma can be achieved in <5 minutes by trained personnel, CT is often difficult to perform if vital signs are unstable, and assessment of other laboratory results is time-consuming. Furthermore, the abovementioned tools are available only at the hospital level, not in the prehospital stage, and require rapid medical decisions. Instead of laboratory information, prehospital paramedics can frequently check DSI to indirectly evaluate shock status and assist triage decisions for trauma centers.

Bruijins et al. reported a significant association between DSI and 48-hour mortality with moderate injury severity [12]. In another study, Bellal et al. included only patients with severe injury (ISS >15) in the analysis showed that the mortality was high in the positive DSI group [10]. The effect of DSI on injury severity is interesting as clinical deterioration is not clear in individuals with moderate injury severity. However, injury severity evaluation requires hospital information such as CT and transfusion amount, which is not available at ED admission point. Therefore, we included patients with injury location to abdomen, chest, extremities, or skin evaluated by medical staffs at ED admission instead of using AIS score. This dfference in patient inclusion process has caused inclusion of minor injuries. The proportion of patients with minor injury was 82.1% and moderate injury was 14.0% in our study (Table 2), while moderate injury was 53% in the study by Bruijins et al. However, even if minor injury population was included in the analysis, the result was similar with previous studies that DSI>0.1 group showed higher mortality [10, 12]. Future studies must be conducted prospectively to determine the effects of DSI on injury severity.

## Limitations

This study has several limitations. First, many prehospital SBP and HR values missing in this study. It is challenging to collect data on prehospital vital signs worldwide [9, 12]. These excluded prehospital missing values could have affected the results. If the prehospital time is long or the prehospital SI is high, the EMS provider may administer the intravenous fluid. We did not analyze prehospital treatment in this study. PATOS clinical research network could explore prehospital vital signs and treatments in future studies. Second, this was a retrospective observational study. There could be a measurement error in assessing SBP or HR in prehospital or ED triage. Furthermore, selection bias may have occurred while including specific patient types. Data collection errors can be inherent in the study design. Third, there was insufficient clinical information after ED admission. Information hospital clinical variables, such as CT findings, operation type, blood transfusion products, underlying disease, and medication, could have influenced the data, but the data were not available from the database. Further studies should include these clinical variables. Fourth, different anatomical injuries were included in the analysis. Chest trauma can cause pneumothorax, which can lead to respiratory deterioration, which also affects mortality and ICU admission. Unlike abdominal injury, where hemostasis from the surface is limited, extremity trauma can be easily assessed and managed with hemostasis. This heterogeneity in the study population requires further evaluation in future studies. Fifth, the level of the treating hospital was not evaluated. Even if the participating hospitals of the PATOS are mainly tertiary teaching hospitals, not all participating hospitals are trauma centers. The effects of trauma centers must be considered in future studies. Finally, this study used an international trauma registry across the Asia-Pacific region. Each participating Asian country has different prehospital and hospital treatment protocols from those of European or North American countries. This variability could have influenced the outcomes, and caution is needed while extrapolating the results in different trauma management settings.

## Conclusion

A positive DSI (ED SI worse than EMS SI) is associated with higher mortality, ICU admission, and massive transfusion. 0.1 can be considered as the cut-off value of DSI. Emergency physicians and related stakeholders must consider early activation of the trauma team, including surgeons or embolization interventionists, or prepare for ICU admission and transfusion in cases of hemorrhagic deterioration. Future prospective studies are required to determine an association between DSI and outcomes.

## Supporting information

**S1 Table. Association between exposure groups and outcomes in multivariate logistic regression.**
(DOCX)

**S2 Table. Association between exposure groups and outcomes according to prehospital shock index in multivariate logistic regression.**
(DOCX)

**S1 Appendix. PATOS clinical research network.**
(DOCX)

## Acknowledgments

On behalf of the PATOS Clinical Research Network, we thank the investigators and researchers of all countries and institutions participating in PATOS research.

## Author Contributions

**Conceptualization:** Dae Kon Kim.

**Data curation:** Kyoung Jun Song, Ki Jeong Hong, Tae Han Kim, Sabariah Faizah Jamaluddin.

**Formal analysis:** Dae Kon Kim, Young Sun Ro.

**Investigation:** Young Sun Ro.

**Methodology:** Dae Kon Kim, Joo Jeong, Young Sun Ro.

**Project administration:** Sang Do Shin.

**Resources:** Sang Do Shin, Kyoung Jun Song, Ki Jeong Hong.

**Software:** Sang Do Shin.

**Supervision:** Joo Jeong, Kyoung Jun Song, Ki Jeong Hong, Young Sun Ro.

**Writing – original draft:** Dae Kon Kim, Joo Jeong.

**Writing – review & editing:** Sang Do Shin, Kyoung Jun Song, Ki Jeong Hong, Young Sun Ro, Tae Han Kim, Sabariah Faizah Jamaluddin.

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
