## [Decision Letter · Decision Letter 0]

16 Jun 2021

PONE-D-21-17211

Association between prehospital field to emergency department delta shock index and in-hospital mortality in patients with torso and extremity trauma: a multinational, observational study

PLOS ONE

Dear Dr. Jeong,

Thank you for submitting your manuscript to PLOS ONE. After careful consideration, we feel that it has merit but does not fully meet PLOS ONE’s publication criteria as it currently stands. Therefore, we invite you to submit a revised version of the manuscript that addresses the points raised during the review process.

We look forward to receiving your revised manuscript.

Kind regards,

Zsolt J. Balogh, MD, PhD, FRACS

Academic Editor

PLOS ONE

Journal Requirements:

Additional Editor Comments (if provided):

Dear Authors,

Your paper needs major revision to be considered again, some senior reviewers even suggested rejection.

The two fundamental things you need to address beyond all the questions and concerns addressed in itemized format are:

1. Make the study pragmatic: use only variables in the model and patient population, which are available as latest on ED arrival...clearly not anatomical scores etc.

2. Please express the additional value of delta SI in the context of other parameters/vital signs available on admission. IS dSI is a better one than those and can be used as a single number better than the rest or still just helpful in the process of complex human pattern recognition.

Reviewers' comments:

Reviewer's Responses to Questions

**Comments to the Author**

1. Is the manuscript technically sound, and do the data support the conclusions?

Reviewer #1: Yes

Reviewer #2: No

Reviewer #3: Yes

2. Has the statistical analysis been performed appropriately and rigorously? 

Reviewer #1: Yes

Reviewer #2: Yes

Reviewer #3: Yes

3. Have the authors made all data underlying the findings in their manuscript fully available?

Reviewer #1: Yes

Reviewer #2: Yes

Reviewer #3: Yes

4. Is the manuscript presented in an intelligible fashion and written in standard English?

Reviewer #1: Yes

Reviewer #2: Yes

Reviewer #3: Yes

5. Review Comments to the Author

Reviewer #1: This retrospective observational study seeks to determine the value of the delta shock index between prehospital and emergency department in patients with thoracic, abdominal and extremity injuries, as a measure of hypovolemia, in relation to in-hospital mortality. It uses data derived from the Pan-Asian Trauma Outcomes Study between 2015 to 2018. The key messages can be summarized with the following: a DSI >0.1 is associated with higher in-hospital mortality, admission into the ICU and embolization rates.

Strengths – The findings in this article add to the value of the DSI as an easily measurable tool in the assessment of injured patients that may require further treatment and hemostasis. It’s interesting that one of the endpoints the author chose was embolization rather than operative means of hemostasis. The data mostly supports the author’s conclusions.

Weakness – As the authors stated, the limitation of this paper is the variability of pre-hospital and hospital management of severe trauma between countries and subdivisions within the countries. The implementation of pre-hospital resuscitation that alters the DSI may also differ in a similar way.

Overall – Quality submission. The findings of DSI as an easily measurable tool will certainly help guide pre-hospital and hospital resuscitation and treatment of patients at risk of hemorrhagic shock.

Comments:

1. Title: No issues

2. Abstract: P2L32-36, authors failed to mention exclusion of certain countries from the PATOS data.

3. Abstract: P2L37, authors can also mention their secondary and tertiary outcomes, which they have drawn their conclusions from.

4. Keywords: They adequately reflect the content of the article.

5. Introduction: Well written and summarized previous research with DSI. The authors highlight the differences between previous research that included TBI and facial/neck injuries, and their research that concentrated in thoracic, abdominal and extremity injuries. Clear, explicit reasons for objectives given, and concise. The results and discussion relate to the hypothesis presented in the introduction.

6. Methods: No issues, well presented

7. Results: P8L153 and Fig 1, suggest using the same terms in the figure and what is written on L153 rather than writing “unknown exposure/outlier”.

8. Results: P12L172-173 unclear sentence “The ICU LOS was longer in the DSI >0.1 group with a median of than in the DSI ≤0.1 group (median [IQR]: 6 [3–12.5] vs. 3 [2–7] days).”

9. Results: Table 5, formatting error row 4.

10. Discussion: The results were appropriately discussed and the conclusions were supported by the results. The authors outlined the limitations of the study well.

11. Conclusion: Reflects the aims of the paper.

Reviewer #2: the authors have addressed an a common issue in the trauma patient, what happens when vital signs worsen. Franklin showed this over two decades ago.. does the current analysis reveal anything different? is delta SI and better than delta SBP or MAP of PP? this is an analysis that the authors should do. I am also concerned with the exclusions based on AIS and ISS. these data are not available in the ED when decisions are made limiting the usefulness of the delta SI when caring for individual patients. the authors also don't describe what was done when a significant delta SI as seen.

Reviewer #3: Thank you for the opportunity to review this paper.

A retrospective interrogation of the Pan Asian Trauma Outcome Study has been performed that was able to include data from three of the countries within the database. In keeping with other studies that have documented the use of the delta SI value, the authors have shown that it can potentially help stratify early mobilisation of attention and resources for trauma patients. I think the work highlights a useful index of trauma severity that can be used in trauma management and thus merits publication.

I do though have several questions and suggestions that require response &/or amendments that would make the paper more readable.

Introduction

line 53 suggest changing “Hemorrhage causes hypovolemic shock due to lactic acidosis, hypo…..”

To …..hypovolemic shock, compounded by lactic acidosis,….

Methods

Line 110 “those with abnormal SBP or HR were excluded from the study”

What does this mean? I would have thought that if anything, these unstable patients should be included in the study

Line 112 “The exposure was defined as DSI, ie the change in the SI” I suggest that this is clarified. Presumably the authors mean that the first EMS SI was subtracted from the ER SI.

Line 125

The tertiary outcome was embolization. It is not clear why surgery was not also listed as an outcome when it probably should be to give a better overall view of the usefulness of the DSI value. At least it should be mentioned as a limitation of the study in the discussion.

Line138

The key issue in this paper is the deterioration, if any, during the time that the patient’s SI value was first recorded by the EMS team and the first recording in the ER. From Table 1, there was a long retrieval time in some cases (48 minutes). And yet, treatment during that period was not mentioned, eg fluids, medication usage such as opiates. Could the authors suggest why this data was not mentioned and would they consider including this in future studies?

Results

Line 149 “those aged >18 years or >85 years” , should read “those aged < 18 years or >85 years”

TABLE 1

a. There is a lot of raw data in table 1 and many percentages that could be omitted to allow for easier viewing. For example, it is not clear why the age groups are divided into two groups (19-65 and 66-85) when the effect of age was not one of the main study aims. I would suggest that, unless it is a major point, the average ages and IQR’s would suffice.

b. Also, it would aid readability by sparing the use of percentages - I don’t think they add a lot when the raw figures are already there. The percentage columns are unnecessarily cumbersome .

c. Also, the p values don’t always seem to match the data sets. For example, for age, the p value (<0.01) presumably should be on the same line as the median values The same applies to the p values for EMS call to ED arrival, EMS SI / HR / SBP and HR.

d. Also, consider leaving out the section on intent, I can’t see how this is relevant to the paper. Mechanism and anatomical location should be adequate.

TABLE 2

a. Again, please consider leaving out the percentages or perhaps putting them in parentheses next to the raw datum – eg 470 (100).

b. The p values should be in the same line as the median averages.

c. Some of the p values don’t make sense. Eg, there were 6 cases where the injury was localised to the chest, with 3 being in the DSI <0.1 group and 3 in the DSI >0,1 group and yet there was a p value of 0.02, is this correct?

TABLE 3

Consider placing the 95%CI in parentheses next to the OR values rather than in separate columns

TABLE 4

a. As with previous comments, consider deleting the percentage values.

b. Again, the p values should be on the same line as the median figures.

TABLE 5

Same suggestion re CI’s as Table 4.

Discussion

Lines 268-270 The paper by Bruijins et al is criticised for not indicating what the appropriate treatment was during the study period. However, this study (as mentioned above for line 138), also did not mention any treatment administered by the EMS teams. This should be noted as a limitation.

Conclusion

Line 303-304 Given that the authors have found higher AORs (Table 5) for mortality, ICU admission and embolization if the DSI is >0.1, would they consider seeking a cut- off value for escalation of treatment, or is >0.1 considered to be the cut-off?

Overall, an interesting study which further highlights the potential value of using the DSI as an adjunct to decision making. With some tidying up, especially of the tables, it should merit publication. Thank you.

6. PLOS authors have the option to publish the peer review history of their article (what does this mean?). If published, this will include your full peer review and any attached files.

Reviewer #1: No

Reviewer #2: No

Reviewer #3: No

---

## [Author Response · Author response to Decision Letter 0]

18 Aug 2021

Author’s reply to the reviewer’s comments:

On behalf of the authors, thank you for the valuable comments by the reviewer of our paper. We have attempted to address every point raised by the reviewer in the revised manuscript. While we believe that we have addressed all of the reviewer's concerns, we would be more than pleased to write additional revisions if necessary.

We have highlighted all of the changes, the authors’ answers, and explanations.

Correspondent author

Additional Editor Comments (if provided):

Dear Authors,

Your paper needs major revision to be considered again, some senior reviewers even suggested rejection.

The two fundamental things you need to address beyond all the questions and concerns addressed in itemized format are:

1. Make the study pragmatic: use only variables in the model and patient population, which are available as latest on ED arrival...clearly not anatomical scores etc.

-> Thank you for your thoughtful comments. As you advised, we reran the analysis to make this study pragmatic. We have completely redefined the study population. Only variables that we can use upon arrival at ED were used in the inclusion criteria and multivariable logistic regression analysis. We excluded anatomical scores in study enrollment. We added this part in response to reviewer #2's comments.

2. Please express the additional value of delta SI in the context of other parameters/vital signs available on admission. IS dSI is a better one than those and can be used as a single number better than the rest or still just helpful in the process of complex human pattern recognition.

-> Thank you for your thoughtful comments. As you advised, we tried to describe whether the delta SI has an additional value-added to other parameters or vital signs. We added this part in response to reviewer #2's comments. 

Reviewer #1: This retrospective observational study seeks to determine the value of the delta shock index between prehospital and emergency department in patients with thoracic, abdominal and extremity injuries, as a measure of hypovolemia, in relation to in-hospital mortality. It uses data derived from the Pan-Asian Trauma Outcomes Study between 2015 to 2018. The key messages can be summarized with the following: a DSI >0.1 is associated with higher in-hospital mortality, admission into the ICU and embolization rates.

Strengths – The findings in this article add to the value of the DSI as an easily measurable tool in the assessment of injured patients that may require further treatment and hemostasis. It’s interesting that one of the endpoints the author chose was embolization rather than operative means of hemostasis. The data mostly supports the author’s conclusions.

Weakness – As the authors stated, the limitation of this paper is the variability of pre-hospital and hospital management of severe trauma between countries and subdivisions within the countries. The implementation of pre-hospital resuscitation that alters the DSI may also differ in a similar way.

Overall – Quality submission. The findings of DSI as an easily measurable tool will certainly help guide pre-hospital and hospital resuscitation and treatment of patients at risk of hemorrhagic shock.

-> Thank you very much for summarizing the key points, strengths, and weaknesses of our research. As you said, the circumstances of the various countries and organizations participating in PATOS are different. Distinguishing these specific differences in detail is not easy in this study. However, we think it is meaningful to analyze data from three Asian countries (Malaysia, South Korea, and Taiwan). We have corrected what you pointed out as much as possible. We have responded to the sentences you recommended to be corrected.

Comments:

1. Title: No issues

2. Abstract: P2L32-36, authors failed to mention exclusion of certain countries from the PATOS data.

-> Thank you for your valuable comment. We further mentioned countries excluded from the PATOS data.

Patients who were transferred from another facility; those who were transferred without the use of emergency medical service; those with prehospital cardiac arrest and severe trauma (Abbreviated Injury Scale score ≥3) in the head, face, neck, spine, or body surface; those with unknown exposure and outcomes were excluded.

-> Patients from China, Indonesia, Japan, Philippines, Thailand, and Vietnam; those who were transferred from another facility; those who were transferred without the use of emergency medical service; those with prehospital cardiac arrest; those with unknown exposure and outcomes were excluded.

3. Abstract: P2L37, authors can also mention their secondary and tertiary outcomes, which they have drawn their conclusions from.

-> Thank you for your valuable comment. We mentioned secondary and tertiary outcomes (We changed the tertiary outcome through revision concerning the opinions of another reviewer). 

The exposure and primary outcome were DSI and in-hospital mortality, respectively.

-> The exposure and primary outcome were DSI and in-hospital mortality, respectively. The secondary and tertiary outcome was intensive care unit (ICU) admission and massive transfusion, respectively.

4. Keywords: They adequately reflect the content of the article.

5. Introduction: Well written and summarized previous research with DSI. The authors highlight the differences between previous research that included TBI and facial/neck injuries, and their research that concentrated in thoracic, abdominal and extremity injuries. Clear, explicit reasons for objectives given, and concise. The results and discussion relate to the hypothesis presented in the introduction.

6. Methods: No issues, well presented

7. Results: P8L153 and Fig 1, suggest using the same terms in the figure and what is written on L153 rather than writing “unknown exposure/outlier”.

-> Thank you for your valuable comment. We have corrected the term of the figure as you said (As pointed out by reviewer #3, the term “abnormal” has been changed to “outlying”).

8. Results: P12L172-173 unclear sentence “The ICU LOS was longer in the DSI >0.1 group with a median of than in the DSI ≤0.1 group (median [IQR]: 6 [3–12.5] vs. 3 [2–7] days).”

-> Thank you for your valuable comment. We have corrected the sentence you pointed out.

The ICU LOS was longer in the DSI >0.1 group with a median of than in the DSI ≤0.1 group (median [IQR]: 6 [3–12.5] vs. 3 [2–7] days).

-> The median ICU LOS was longer in the DSI >0.1 group than in the DSI ≤0.1 group (median [IQR]: 5 [2–11] vs. 3 [2–8] days).

9. Results: Table 5, formatting error row 4.

-> Thank you for your valuable comment. We fixed it by adjusting the column spacing.

10. Discussion: The results were appropriately discussed and the conclusions were supported by the results. The authors outlined the limitations of the study well.

11. Conclusion: Reflects the aims of the paper.

 

Reviewer #2: the authors have addressed an a common issue in the trauma patient, what happens when vital signs worsen. Franklin showed this over two decades ago.. does the current analysis reveal anything different? is delta SI and better than delta SBP or MAP of PP? this is an analysis that the authors should do. I am also concerned with the exclusions based on AIS and ISS. these data are not available in the ED when decisions are made limiting the usefulness of the delta SI when caring for individual patients. the authors also don't describe what was done when a significant delta SI as seen.

-> Thank you for the critical and constructive comments. It can be challenging to claim that our study has presented a complete something new. However, we think this study is different from other studies. It was conducted in a multi-center and multi-country, used prehospital and ED shock index, and measured various outcome variables. If we use delta SBP, MAP, or PP rather than delta shock index, it will be a very attractive study. However, we made use of the delta shock index as the main topic of this study. We hope you understand that changing a key variable is very difficult. 

We fully agree with your point that AIS and ISS are not available variables to use at the time of an emergency department visit. ISS was excluded from the inclusion and exclusion criteria. We redefined the study subject by using the injury site. All analyzes were performed anew. Only variables that we can use upon arrival at ED were used in the inclusion criteria and multivariable logistic regression analysis. 

We analyzed this study retrospectively. We do not know whether the delta shock index was actually used when treating patients in the institution's emergency department participating in the data collection. Therefore, it is impossible to accurately show which treatment was performed in patients with a significant delta shock index. However, patients with high delta SI were likely judged to be more likely to have active bleeding. Therefore, transfusion may be the essential treatment in these situations. We changed the tertiary outcome to transfusion. The analysis of embolization was presented as a supplementary table. In addition, additional analysis on surgery was performed and presented as a supplementary table.

 

Reviewer #3: Thank you for the opportunity to review this paper.

A retrospective interrogation of the Pan Asian Trauma Outcome Study has been performed that was able to include data from three of the countries within the database. In keeping with other studies that have documented the use of the delta SI value, the authors have shown that it can potentially help stratify early mobilisation of attention and resources for trauma patients. I think the work highlights a useful index of trauma severity that can be used in trauma management and thus merits publication.

I do though have several questions and suggestions that require response &/or amendments that would make the paper more readable.

-> Thank you very much for commenting on the merits of our study. We have reviewed all the points you pointed out and have revised them as follows.

Introduction

line 53 suggest changing “Hemorrhage causes hypovolemic shock due to lactic acidosis, hypo…..”

To …..hypovolemic shock, compounded by lactic acidosis,….

-> Thank you for your valuable comment. We corrected it, as you pointed out.

Hemorrhage causes hypovolemic shock due to lactic acidosis, hypothermia, and coagulopathy.

-> Hemorrhage causes hypovolemic shock, compounded by lactic acidosis, hypothermia, and coagulopathy.

Methods

Line 110 “those with abnormal SBP or HR were excluded from the study”

What does this mean? I would have thought that if anything, these unstable patients should be included in the study

-> Thank you for your valuable comment. The term “abnormal” has been changed to “outlying” (e.g., when blood pressure is greater than 300 mmHg). 

those with abnormal SBP or HR; and those with unknown outcomes were excluded.

-> those with outlying SBP or HR; and those with unknown outcomes were excluded.

Line 112 “The exposure was defined as DSI, ie the change in the SI” I suggest that this is clarified. Presumably the authors mean that the first EMS SI was subtracted from the ER SI.

-> Thank you for your valuable comment. We corrected it, as you pointed out.

The exposure was defined as DSI, i.e., the change in SI from the prehospital field to the ED.

-> The exposure was defined as DSI, i.e., the first EMS SI was subtracted from the ED SI.

Line 125

The tertiary outcome was embolization. It is not clear why surgery was not also listed as an outcome when it probably should be to give a better overall view of the usefulness of the DSI value. At least it should be mentioned as a limitation of the study in the discussion.

-> Thank you for your valuable comment. As you gave advice, we also performed an analysis for surgery and added it. You can see the results in tables 2 and 4 and supplementary tables 1 and 2.

Line138

The key issue in this paper is the deterioration, if any, during the time that the patient’s SI value was first recorded by the EMS team and the first recording in the ER. From Table 1, there was a long retrieval time in some cases (48 minutes). And yet, treatment during that period was not mentioned, eg fluids, medication usage such as opiates. Could the authors suggest why this data was not mentioned and would they consider including this in future studies?

-> Thank you for your valuable comment. We are collecting data related to prehospital treatment in the PATOS database. However, this study focused on the prehospital vital sign as an independent variable. Treatment at the prehospital stage is essential. We will plan a follow-up study including this part. We described this in the limitation section as follows.

This study has several limitations. First, many prehospital SBP and HR values missing in this study. It is challenging to collect data on prehospital vital signs worldwide [9,12]. These excluded prehospital missing values could have affected the results.

-> This study has several limitations. First, many prehospital SBP and HR values missing in this study. It is challenging to collect data on prehospital vital signs worldwide [9,12]. These excluded prehospital missing values could have affected the results. If the prehospital time is long or the prehospital SI is high, the EMS provider may administer the intravenous fluid. We did not analyze prehospital treatment in this study. PATOS clinical research network could explore prehospital vital signs and treatments in future studies.

Results

Line 149 “those aged >18 years or >85 years” , should read “those aged < 18 years or >85 years”

-> Thank you for your detailed comment. We corrected it, as you pointed out.

those aged >18 years or >85 years (N = 8,829),

-> those aged <18 years or >85 years (N = 8,829),

TABLE 1

a. There is a lot of raw data in table 1 and many percentages that could be omitted to allow for easier viewing. For example, it is not clear why the age groups are divided into two groups (19-65 and 66-85) when the effect of age was not one of the main study aims. I would suggest that, unless it is a major point, the average ages and IQR’s would suffice.

-> Thank you for your valuable comment. If category and median (IQR) were presented together, only median (IQR) was left.

b. Also, it would aid readability by sparing the use of percentages - I don’t think they add a lot when the raw figures are already there. The percentage columns are unnecessarily cumbersome .

-> Thank you for your valuable comment. We put them in parentheses next to the raw datum.

c. Also, the p values don’t always seem to match the data sets. For example, for age, the p value (<0.01) presumably should be on the same line as the median values The same applies to the p values for EMS call to ED arrival, EMS SI / HR / SBP and HR.

-> Thank you for your valuable comment. The p-values of variables tested with the chi-square test and Wilcoxon rank-sum test were calculated to be less than 0.01, which was presented on the top line. We have corrected it, as you pointed out.

d. Also, consider leaving out the section on intent, I can’t see how this is relevant to the paper. Mechanism and anatomical location should be adequate.

-> Thank you for your valuable comment. The intent has been removed from the table.

TABLE 2

a. Again, please consider leaving out the percentages or perhaps putting them in parentheses next to the raw datum – eg 470 (100).

-> Thank you for your valuable comment. We put them in parentheses next to the raw datum.

b. The p values should be in the same line as the median averages.

-> Thank you for your valuable comment. We put the p values in the same lines.

c. Some of the p values don’t make sense. Eg, there were 6 cases where the injury was localised to the chest, with 3 being in the DSI <0.1 group and 3 in the DSI >0,1 group and yet there was a p value of 0.02, is this correct?

TABLE 3

Consider placing the 95%CI in parentheses next to the OR values rather than in separate columns

-> Thank you for your valuable comment. We put them in parentheses next to the raw datum.

TABLE 4

a. As with previous comments, consider deleting the percentage values.

-> Thank you for your valuable comment. We put them in parentheses next to the raw datum.

b. Again, the p values should be on the same line as the median figures.

-> Thank you for your valuable comment. We put the p values in the same lines.

TABLE 5

Same suggestion re CI’s as Table 4.

-> Thank you for your valuable comment. We put them in parentheses next to the raw datum.

Discussion

Lines 268-270 The paper by Bruijins et al is criticised for not indicating what the appropriate treatment was during the study period. However, this study (as mentioned above for line 138), also did not mention any treatment administered by the EMS teams. This should be noted as a limitation.

-> Thank you for your valuable comment. As answered above, we have specified this in the limitation.

This study has several limitations. First, many prehospital SBP and HR values missing in this study. It is challenging to collect data on prehospital vital signs worldwide [9,12]. These excluded prehospital missing values could have affected the results.

-> This study has several limitations. First, many prehospital SBP and HR values missing in this study. It is challenging to collect data on prehospital vital signs worldwide [9,12]. These excluded prehospital missing values could have affected the results. If the prehospital time is long or the prehospital SI is high, the EMS provider may administer the intravenous fluid. We did not analyze prehospital treatment in this study. PATOS clinical research network could explore prehospital vital signs and treatments in future studies.

Conclusion

Line 303-304 Given that the authors have found higher AORs (Table 5) for mortality, ICU admission and embolization if the DSI is >0.1, would they consider seeking a cut- off value for escalation of treatment, or is >0.1 considered to be the cut-off?

-> Thank you for your valuable comment. We considered DSI > 0.1 as a cut-off. However, various cut-off values can be regarded as for escalation of treatment. Therefore, we have revised the first sentence of the conclusion as follows.

DSI >0.1 is associated with a higher rate of mortality, ICU admission, and massive transfusion. 

-> A positive DSI (ED SI worse than EMS SI) is associated with higher mortality, ICU admission, and massive transfusion. 0.1 can be considered as the cut-off value of DSI.

Overall, an interesting study which further highlights the potential value of using the DSI as an adjunct to decision making. With some tidying up, especially of the tables, it should merit publication. Thank you.

-> Thank you for your attentive comments on our entire manuscript. Everything you said has been checked and corrected.

---

## [Decision Letter · Decision Letter 1]

6 Oct 2021

Association between prehospital field to emergency department delta shock index and in-hospital mortality in patients with torso and extremity trauma: a multinational, observational study

PONE-D-21-17211R1

Dear Dr. Jeong,

We’re pleased to inform you that your manuscript has been judged scientifically suitable for publication and will be formally accepted for publication once it meets all outstanding technical requirements.

Kind regards,

Zsolt J. Balogh, MD, PhD, FRACS

Academic Editor

PLOS ONE

Additional Editor Comments (optional):

Reviewers' comments:

Reviewer's Responses to Questions

**Comments to the Author**

1. If the authors have adequately addressed your comments raised in a previous round of review and you feel that this manuscript is now acceptable for publication, you may indicate that here to bypass the “Comments to the Author” section, enter your conflict of interest statement in the “Confidential to Editor” section, and submit your "Accept" recommendation.

Reviewer #1: All comments have been addressed

Reviewer #2: All comments have been addressed

Reviewer #3: All comments have been addressed

2. Is the manuscript technically sound, and do the data support the conclusions?

Reviewer #1: Yes

Reviewer #2: (No Response)

Reviewer #3: Yes

3. Has the statistical analysis been performed appropriately and rigorously? 

Reviewer #1: Yes

Reviewer #2: (No Response)

Reviewer #3: Yes

4. Have the authors made all data underlying the findings in their manuscript fully available?

Reviewer #1: Yes

Reviewer #2: (No Response)

Reviewer #3: Yes

5. Is the manuscript presented in an intelligible fashion and written in standard English?

Reviewer #1: Yes

Reviewer #2: (No Response)

Reviewer #3: Yes

6. Review Comments to the Author

Reviewer #1: (No Response)

Reviewer #2: (No Response)

Reviewer #3: I thank the authors for attending to the issues I raised. I am satisfied with the changes made to the article and support its acceptance for publication.

7. PLOS authors have the option to publish the peer review history of their article (what does this mean?). If published, this will include your full peer review and any attached files.

Reviewer #1: No

Reviewer #2: No

Reviewer #3: No

---

## [Editor Report · Acceptance letter]

15 Oct 2021

PONE-D-21-17211R1 

Association between prehospital field to emergency department delta shock index and in-hospital mortality in patients with torso and extremity trauma: a multinational, observational study 

Dear Dr. Jeong:

I'm pleased to inform you that your manuscript has been deemed suitable for publication in PLOS ONE. Congratulations! Your manuscript is now with our production department. 

Kind regards, 

on behalf of

Dr. Zsolt J. Balogh 

Academic Editor

PLOS ONE